# Modest google trends associations with mental health patient and general population deaths by suicide: A time-series analysis using patient-provided search terms

**Lana Bojanić** ⓘ*, **Isabelle M. Hunt, Sandra Flynn, Pauline Turnbull, Saied Ibrahim**

National Confidential Inquiry into Suicide and Safety in Mental Health, Centre for Mental Health and Safety, Division of Psychology and Mental Health, School of Health Sciences, University of Manchester, Manchester, United Kingdom of Great Britain and Northern Ireland

* lana.bojanic-2@manchester.ac.uk

## Abstract

### Background

Suicide-related internet use (SRIU), particularly online information seeking, seems to be highly prevalent among mental health patients. Most research using Google Trends to examine suicide-related search patterns relies on researcher-selected terms rather than those actually used by people at risk, limiting ecological validity. No studies have specifically examined associations between volume of suicide-related searches and number of suicides by mental health patients, despite this population's elevated risk and prevalent SRIU.

### Methods

This study analysed suicide-related search terms provided by 196 people who had used mental health services in the UK who had engaged in SRIU. From 520 search entries, 36 terms were identified across seven categories and analysed using Google Trends data (2011–2022) for England, Wales, and Scotland. Transfer function modelling with ARIMA/SARIMA models examined cross-correlations between monthly search volumes and the number of suicide deaths by both mental health patients and the general population.

### Results

Of 448 tested lag correlations, 18 (2.0%) were statistically significant, showing small to medium effect sizes. Most significant associations involved non-specific suicide-related queries (33.3%), characteristics of suicide (22.2%), and help-seeking queries (22.2%). No pro-suicide queries showed significant associations. Concurrent positive correlations were found between search volumes for suicide ideation terms and suicide

⬛ OPEN ACCESS

which permits unrestricted use, distribution, and reproduction in any medium, provided the original author and source are credited.

**Data availability statement:** Search term data: The raw participant-level data for this study cannot be made publicly available. Our ethics approval from the University of Manchester Research Ethics Committee 1 (ref: 2023-16133-28055) explicitly requires that: • Data may only be accessed and analysed within the NCISH office and are stored on an isolated network not connected to general university infrastructure. • Access is restricted to the named research team (LB, IH, SI, SF, PT) involved in this study. These restrictions were imposed as a condition of the ethical approval due to the sensitive nature of the data (mental health patients with suicidal thoughts/behaviours). Making these data publicly available would breach both our approved protocol and UK GDPR/Data Protection Act 2018 obligations. Data requests and inquiries may be sent to: research.ethics@manchester.ac.uk. The search volume data is fully dependent on the participant-generated search term data; without the exact search terms, exact Google Trends data can not be replicated. Therefore, providing access to it would not enable full replication of our analysis. Replication would require access to the search term dataset, which is restricted under our ethics approval. Suicide mortality data: Suicide mortality contain potentially identifying and, in case of patient suicide data, sensitive patient information. These data are governed by strict confidentiality regulations and are not available for public distribution. Further, these data are owned by third party organisations. Access to the Office for National Statistics and National Records of Scotland data is subject to request. For more information, see https://www.ons.gov.uk/aboutus/whatwedo/statistics/requestingstatistics (ONS) and https://www.researchdata.scot/accessing-data/begin-an-enquiry (NRS). Access to mental health patient suicide data can be requested via application to the Healthcare Quality Improvement Partnership (https://www.hqip.org.uk/national-programmes/accessing-ncapop-data/).

**Funding:** The author(s) received no specific funding for this work.

**Competing interests:** The authors have declared that no competing interests exist.

deaths by mental health patients, and between help-seeking terms and suicide deaths in the general population. Some search terms, including those related to self-poisoning methods and major suicide prevention charities, showed protective associations, with increased searches preceding decreased suicide deaths by 2–3 months.

## Discussion

The study reveals modest associations between suicide-related search patterns and actual suicide deaths. The associations differed between mental health patients and the general population. The absence of pro-suicide query associations may reflect effective online prevention efforts in the UK. However, the small number of significant correlations indicates limited predictive utility for population-level suicide monitoring, supporting conclusions that Google Trends data alone is insufficient for predicting trends in suicide, though valuable for generating hypotheses about suicide-related internet behaviours.

## Introduction

Suicide is a significant public health issue worldwide, with far-reaching impacts on individuals, families, and communities. In the digital age, a notable behaviour associated with suicide risk is suicide-related internet use (SRIU), which encompasses a range of online behaviours. The most common of these behaviours is suicide-related online information seeking, where individuals search for information about suicide online. This behaviour reflects the complex motivations underlying SRIU: individuals may seek guidance to manage their distress, look for help and support, or, in some cases, actively contemplate suicide. Previous survey-based research has found that the prevalence of suicide-related online information seeking ranges from 66% in the general population to 84% in clinical populations [1–4]. Additionally, recent findings confirm that suicide-related online information seeking was the most common type of SRIU among mental health patients, regardless of whether the motivation was help-seeking or intent to die [1]. This underscores the duality of suicide-related online information seeking, namely; that both the purpose and the outcome of this behaviour can be both beneficial and harmful. Consequently, embedding interventions within search engines presents a promising strategy to enhance suicide prevention efforts at the population level.

Most existing research on suicide-related online information seeking has relied on population-level analyses using tools such as Google Trends. Google Trends allow researchers to estimate the relative frequency of Google searches for terms of interest over time and across regions. Some studies have reported meaningful associations between suicide-related search volumes as well as broader terms connected to suicide risk and suicide mortality or crisis behaviours [5–8]. However, the findings are highly variable and subject to numerous methodological limitations [9]. A fundamental limitation of this body of research is the lack of input from the lived experience of suicidal individuals. Suicide-related search terms analysed in Google Trends studies are

typically selected a priori by researchers, raising concerns about the ecological validity of the findings. Without knowing whether individuals experiencing suicidal ideation actually use these terms, there is a risk of wasting resources researching irrelevant searches and overlooking relevant terms. Moreover, a growing trend in Google Trends research involves testing dozens or even hundreds of search terms without a solid theoretical or empirical rationale; this approach increases the risk of false positives and undermines the interpretability of results [6,7,10].

No existing Google Trends studies on suicidality have focused specifically on suicides among mental health patients as the primary outcome. Instead, the majority of Google Trends-based research has examined general population suicide trends, overlooking the unique characteristics and online behaviours of individuals in contact with mental health services. This represents a critical gap, as mental health patients constitute a high-risk group whose patterns of internet use, familiarity with mental health terminology, and help-seeking behaviours may diverge significantly from those of the general public [3]. Exploring internet searches relating to suicide specifically within this clinical population could yield more precise and actionable insights for targeted prevention strategies. To address these gaps, this study aims to [1] explore patterns of online suicide-related information seeking using search terms provided by mental health patients and [2] examine whether these patterns are associated with actual suicides among mental health patients and the general population.

## Methods

### Participants who provided search terms for analysis

The participants were all aged 18 or over, lived in the UK, had been in contact with mental health services in the preceding year and had experienced suicidal thoughts or behaviours and engaged in SRIU during this time. Characteristics of the full sample are described in detail elsewhere [1]. Participants were asked to write down the search terms (queries) they had used when seeking suicide-related content online [1]. Of 533 participants who engaged in suicide-related internet use, 196 (37%) provided their search terms. This group included 10 participants who reported encountering suicide-related content accidentally but still provided search terms. Overall, participants provided 520 usable entries. On average, each participant contributed 2.6 search terms (range = 1–25), and each search term was treated as an independent unit of analysis. Characteristics of participants who provided search terms were examined.

### Search term selection

The process of selecting search terms for extracting Google Trends data involved both qualitative and quantitative approaches, which comprised the following steps:

1. Compilation and coding of search terms

    1.1. Firstly, search terms were ordered by frequency. Those that were used by at least two participants were designated base search terms. Three of these base terms were excluded as they were not related to the topic of suicide.

    1.2. All search terms (base search terms and others) were coded and organised into seven broad categories: a) non-specific suicide-related queries, b) symptoms or diagnosis, c) seeking help queries, d) pro-suicide queries, e) suicide methods, f) procedural queries, and g) characteristics of suicide (see Table 1 in Results section). The coding was carried out by LB and reviewed by SF and SI.

2. Identification of synonyms and related phrases

Synonyms, abbreviations, alternative spellings, and related phrases for base search terms 1.1.) were searched for within their categories 1.2.). These were then combined into finalised search terms using Google Trends Boolean operators (i.e., '+' for combining and '−' for excluding terms). For example, for the base search term 'overdose' (searched for by

**Table 1. Categories of search terms and their descriptions, including the number of unique search terms and the frequency with which participants reported those terms.**

| Type of search term | Description | N unique search terms extracted | N instances reported by participants |
|---|---|---|---|
| Suicide methods | Queries relating either to a suicide method in general or to one of its aspects, such as lethality or accessibility | 11 | 90 |
| Non-specific suicide-related queries | Broad references to suicidality, excluding specific descriptions, methods or intentions | 6 | 127 |
| Seeking help | Explicit help-seeking motivated terms | 6 | 24 |
| Pro-suicide queries | Known pro-suicide and pro-euthanasia websites, books or forums | 5 | 32 |
| Characteristic of suicide | Specific attributes of suicide; such as desired physical or emotional qualities. | 5 | 22 |
| Procedural queries | Questions seeking guidance on how to carry out suicidal behaviour ("how-to" phrasing) | 2 | 15 |
| Symptoms or diagnosis | Inquiries about signs of mental illness | 1 | 2 |

9 participants), its category e) (suicide methods) was explored for related content and identified two participants who searched for the abbreviation 'od'. This additional search term was combined with the base search term as follows: 'overdose + od'. This was carried out by LB with input from SI and SF.

3. Search term optimisation via related topics and related queries features

The finalised search terms were entered into the Google Trends website to examine the results from the "related topics" and "related queries" features, which provide insight into topics, phrases, or questions commonly searched alongside the selected terms [11]. In some cases, search terms were revised based on this examination; for example, 'toxicity' was modified to 'toxicity -dog' to exclude irrelevant results related to common searches for the toxicity of substances for dogs. The final number of search terms for which data was extracted was 36.

## Obtaining and preparing Google Trends data

Monthly Google Trends data for 36 selected search terms were collected for England, Wales, and Scotland, covering the period from January 1, 2011, to December 31, 2022. Google Trends data was not obtained for the whole of the UK as suicide mortality data was not available on the level of month for Northern Ireland (see next paragraph). The start year was chosen because it marks when over 85% of the UK population was using the internet and Google held a 92% search engine market share, indicating that from this point onward, Google Trends data would be sufficiently representative of UK online activity [12,13]. The final year, 2022, was selected as it was the most recent year with comprehensive mental health patient suicide statistics available [14]. Steps were taken to address potential bias introduced by the Google Trends sampling process. Namely, Google Trends time series for any given search term are based on a sample rather than the whole search volume, and each sample somewhat differs from another, raising concerns of replicability [15]. To address this, the data were retrieved on 15 separate days in March 2025 and averaged to reduce random fluctuations, following the recommendations of Medeiros and Pires [15].

Before averaging, the obtained data were normalised in three steps: firstly, for each search term, the values for each devolved nation were weighted by the respective population percentage of each nation, producing a composite index. Population percentages were determined using yearly population estimates for each year from 2011 to 2022. Weighting Google Trends data by population size was appropriate given the broadly similar internet infrastructure across the UK [16], online searching behaviour [17], and mental health related help-seeking practices [18].

Secondly, a maximum value for each search term was identified. Finally, the composite index value was divided by this maximum value and multiplied by 100 to restore the traditional 0–100 scale characteristic of Google Trends data. After normalisation, data were averaged, creating 36 unique Google Trends time series. Time series with more than 15% data points missing and/or 4 consecutive months having missing data were not analysed. After the pattern of missingness was analysed, four search terms were removed due to having more than 15% of missing data (1 seeking help, 1 suicide method, 1 pro-suicide and 1 of a non-specific suicide-related category). Time series with 0.1 to 15% missing data points were imputed using linear interpolation [19,20]. This was the case in search volumes for nine other search terms (28.1%).

## Mental health patient and general population suicide time-series

Actual monthly numbers of mental health patients who died by suicide were obtained from the National Confidential Inquiry into Suicide and Safety in Mental Health (NCISH). Note that the more detailed NCISH methodology is available elsewhere [21]. NCISH collects data on mental health patients who die by suicide in the UK through a three-stage process. Firstly, the national mortality data on all deaths classified as suicide or undetermined intent are collected from the national statistical bodies of the UK's devolved nations. Secondly, individuals who had contact with specialist mental health services within 12 months prior to death are identified via the healthcare providers in the area where the person lived and died. Finally, detailed clinical and socio-demographic information is collected via questionnaires completed by the patient's supervising clinician. Note that NCISH cases are simply a subset of ONS/NRS cases; ONS/NRS establishes whether a death is a suicide, NCISH then establishes mental health service contact for those cases. Due to the lower data completeness in recent years caused by the time associated with legal processes, identifying patient cases, and obtaining information via questionnaires, monthly data for the years 2020, 2021 and 2022 for England and 2021 and 2022 for Wales and Scotland were projected. These projections were based on the number of unreturned questionnaires, the rate of returns in previous years, and the accuracy of estimates in previous years. This is a standard process for reporting patient suicide figures from recent years [14]. Due to the data on the month of death not being available for Northern Ireland, only data for England, Wales and Scotland were analysed. Monthly general population suicide data is routinely provided to NCISH by the Office for National Statistics and National Records Scotland. Rather than rates, actual suicide counts were used in the analysis since the suicide data already represented complete Great British population of suicide deaths, both in general population and in mental health patients. Notably, suicide counts and suicide rates are highly correlated in our dataset (r = 0.97). Therefore, the correlation between the weighted GT composite and absolute suicide counts result in the same statistical inference to what we would obtain using rates. Data were accessed for research purposes on the 1st of March 2025. Authors had no access to information that could identify individual participants during or after data collection.

## Statistical analysis

Characteristics of participants who provided search terms have been described using frequencies and proportions (for categorical) and median and interquartile range (IQR) (for continuous variables, visual analogue scales). Chi-square tests were used for comparisons of categorical variables, while the Wilcoxon test was employed for comparisons of visual analogue scale data between participants who did and those who did not provide search terms. Associations between Google Trends and mental health patient deaths by suicide time-series were determined using transfer function modelling, as recommended by Tran et al [9]. This was done on both on both full (2011−2022) and sensitivity time period (2011−2020). In the first step, explanatory time-series (Google Trends time-series) was modelled using autoregressive integrated moving average (ARIMA) models; if there was a seasonal component, seasonal ARIMA (SARIMA) was used. Modelling included explanatory time-series outliers, if present. Types of outliers considered were innovational outliers (IO), additive outliers (AO), level shifts (LS), temporary changes (TC) and seasonal level shifts (SLS) [22]. This modelling was iterative, and adequate models were determined based on a) corrected Akaike Information Criterion (AICc) and Bayesian Information Criterion (preferring models with comparatively lower values), b) inspection of the autocorrelation functions (ACF) and

the partial autocorrelation functions (PACF) of the model, and c) testing for the presence of remaining autocorrelation of residuals using Ljung-Box statistics. The identified adequate model was then fitted to dependent time series (time series of mental health patient/general population suicide). Finally, the cross-correlation function (CCF) was used to determine the correlation between prewhitened explanatory and dependent time series. CCF was examined for lags from −3 to +3. In order to correct for multiple testing, the Bonferroni correction was applied (p < 0.05/number of tests). All analyses were carried out in RStudio [23], with Google Trends obtained using package gtrendsR [24] and analyses carried out using packages tsoutliers [25] and forecast [26].

### Ethical approval

Ethical approval for collecting and analysing participants' suicide-related search terms was obtained from the University of Manchester Research Ethics Committee 1 on the 3rd of April 2023 (ref: 2023-16133-28055). Suicide data is routinely obtained by the National Confidential Inquiry into Suicide and Safety in Mental Health with approval from the North West – GM South REC (reference: ERP/96/136) and Section 251 Approval under the NHS Act 2006 (originally Section 60 of the Health and Social Care Act 2001) (reference: 23/CAG/0024) for England and Wales and Public Benefit and Privacy Panel for Health and Social Care (HSC-PBPP) (reference: 2021−0114) for Scotland.

## Results

### Characteristics of participants who provided search terms

Participants who provided search terms were mostly aged between 18 and 34 years (129, 65.9%), female (133, 67.9%) and White (164, 83.7%). Over a third were lesbian, bisexual, or gay (72, 36.7%) and a further 13.3% [26] did not identify with the sex they were assigned at birth. Over half were employed (105, 53.6%), and 61.7% (121 participants) had a university degree or higher qualification. The most common diagnoses were affective disorder (139, 70.9%) and anxiety disorder (133, 57.7%), and 158 (80.6%) had been prescribed psychotropic medication. Participants who provided search terms did not differ from other participants who engaged in SRIU on sociodemographic or clinical characteristics.

Participants who provided search terms most commonly experienced thoughts of suicide every day or almost every day (78, 39.8%) and 89.3% (175) disclosed their thoughts of suicide to someone offline (i.e., a mental health professional, friend, family member). A similar proportion of those who did and who did not provide search terms had attempted suicide in the last year (33.2% v. 31.5%), but those who had provided search terms reported more planning involved in their suicide attempt (median = 7.7, IQR = 5.0–9.0 v. median = 6.0, IQR = 4.0–8.0, p = 0.002).

Participants who had provided search terms mostly searched online for suicide-related content/information (174, 88.8%) and used the internet to interact/connect with others (99, 50.5%), similar to participants who did not provide search terms. The most common motivations to engage in SRIU were seeking help (107, 54.6%) and searching for methods of suicide; the latter was more common for those who provided search terms than in other patients who engaged in SRIU (136, 69.4% v. 182, 54.0%, p = 0.001). Furthermore, 26% [51] of participants who provided search terms were searching for a specific suicide-related site and 66% (130) had engaged in SRIU at least once a month. While engaging in SRIU, the majority (171, 87.2%) had seen suicide prevention messages. Nearly half (83, 48.5%) who had seen prevention messages engaged with them (i.e., called a helpline or visited an advertised prevention website). Finally, participants who provided search terms found their SRIU overall neither helpful nor harmful (median = 5.0, IQR = 4.0–6.4), similar to other participants who had engaged in SRIU. Full comparison between participants who did and those who did not provide search terms is available in Tables 1–4 in S1 File

### Characteristics of search terms

Table 1 categorises the suicide-related search terms reported by participants, including the number of unique search terms extracted per seven categories and the frequency with which participants reported those terms. Most of the unique

search terms extracted were categorised as suicide methods-related (11, 30.6%). Participants mostly reported searching for non-specific suicide-related queries (127, 40.7%), followed by terms related to suicide methods (90, 28.8%).

## Models and cross-correlation coefficients

Table 2 shows the type of SARIMA or ARIMA models that were fitted to explanatory time-series (Google Trends data) for which cross-correlation coefficients on at least one lag were significant. ARIMA models account for time-dependent patterns in the data, while SARIMA models additionally model recurring seasonal effects. The exact search terms are not provided in the table due to ethical considerations regarding the public dissemination of potentially harmful search terms related to suicidality (see [27]).

Of 448 lags (lags from −3 to +3 for each time series) whose cross-correlation coefficient significance was tested, only 18 (2.0%) were significant for both mental health patient and general population suicide. Overall, these cross-correlational coefficients were of medium ($.30 \le |r| < .50$, 22%) and small ($0.10 \le |r| < 0.30$, 78%) effect sizes. Most of the significant coefficients were present for categories of non-specific suicide-related queries (6/18, 33.3%), followed by characteristics of

**Table 2. Fitted Models of Time Series Data with Significant Cross-Correlation Coefficients.**

| Broad category of search term | Description of search term | Full period (2011–2022) | | | Sensitivity period (2011–2020) | | |
|---|---|---|---|---|---|---|---|
| | | Model fitted | Patients* | General population* | Model fitted | Patients* | General population* |
| Non-specific suicide-related query | Expression of intent to die by suicide | SARIMA(1,0,1)(2,1,0)$_{12}$[a] | / | / | SARIMA(0,1,1)(1,0,0)$_{12}$[a] | −3 (−0.25) | −3 (−0.31) |
| Characteristic of suicide | Minimum pain suicide | SARIMA(1,0,1)(0,1,1)$_{12}$ | 0 (0.26) | / | SARIMA(1,0,1)(2,1,1)$_{12}$[b] | 0 (0.25) | 0 (0.25) |
| Characteristic of suicide | Pain-free suicide | SARIMA(1,0,1)(2,1,0)$_{12}$[a] | / | 0 (0.25) | SARIMA(1,0,1)(2,1,0)$_{12}$ | / | / |
| Procedural query | 'How to' general suicide query | SARIMA(0,0,1)(0,1,1)$_{12}$[a] | / | / | SARIMA(1,0,0)(2,1,0)$_{12}$[a] | 0 (0.24) | / |
| Seeking help query | Expression of desire to stop being suicidal | ARIMA(0,1,1) [a] | / | / | ARIMA(0,1,1) [a] | / | +1 (0.27) |
| Symptom or diagnosis | Mentions of anxiety | SARIMA(2,1,2)(1,0,0)$_{12}$[a] | / | / | SARIMA(4,1,0)(0,1,1)$_{12}$[a] | 0 (0.27) | / |
| Suicide method | References hanging | SARIMA(0,0,4)(0,1,1)$_{12}$[a] | / | 0 (0.24) | SARIMA(0,0,0)(0,1,1)$_{12}$[a] | / | / |
| Suicide method | References self-poisoning | SARIMA(0,1,4)(1,1,0)$_{12}$[a] | −2 (−0.24) | / | SARIMA(0,1,4)(1,1,0)$_{12}$[a] | / | / |
| Seeking help query | Suicide prevention charity | SARIMA(1,0,0)(0,1,1)$_{12}$[a] | / | / | SARIMA(1,0,0)(0,1,1)$_{12}$[a] | −3 (−0.25) | / |
| Non-specific suicide-related query | Generic suicidal ideation | SARIMA(1,1,1)(0,1,1)$_{12}$[a] | 0 (0.32) | / | SARIMA(0,1,1)(1,1,1)$_{12}$[a] | 0 (0.38) | / |
| Non-specific suicide-related query | Generic suicide term | ARIMA(0,1,2) [a] | / | +2 (0.27) | SARIMA(0,1,1)(0,1,1)$_{12}$[a] | / | / |
| Non-specific suicide-related query | Generic suicide methods | SARIMA(0,0,2)(1,1,1)$_{12}$[a] | / | / | SARIMA(3,0,0)(1,1,0)$_{12}$[a] | / | +1 (0.25) |
| Seeking help query | Generic suicide support | SARIMA(0,1,2)(0,1,1)$_{12}$[a] | / | 0 (0.30) | SARIMA(0,1,2)(2,1,0)$_{12}$[a] | / | 0 (0.25) |

*$p < 0.002$.

[a]Model fitted including outliers.

[b]Model includes average period-to-period change (drift).

suicide (4/18, 22.2%) and seeking help queries (4/18, 22.2%). Notably, none of the pro-suicide queries yielded significant cross-corelations with actual suicide time series.

Over the full period, there were two significant concurrent (lag 0) positive cross-correlations between search volumes and deaths by suicide of mental health patients time-series; one for a term related to a characteristic of suicide (minimum pain suicide, 0.26) and one for a non-specific suicide-related term (generic suicidal ideation, 0.32). This indicates that increases in mental health patient deaths co-occurred with increases in search volume for these terms. Additionally, there was a significant negative cross-correlation at lag −2 for a suicide method-related search term (referencing self-poisoning, −0.24), suggesting that an increase in search volume for this term was followed by a decrease in mental health patient deaths two months later. Over the full period, there were three concurrent positive cross-correlations between suicide-related search terms time series and general population suicides; one each for search volumes related to a characteristic of suicide (pain-free suicide, 0.25), suicide method (referencing hanging, 0.24), and seeking help (generic suicide support, 0.30). An additional positive cross-correlation was significant at lag 2 (generic suicide term, 0.27).

Over the shortened (sensitivity) period (2011–2020), there were four concurrent significant positive cross-correlations; an increase in search volume for non-specific suicide-related queries (generic suicidal ideation search term, 0.38), characteristics of suicide (minimum pain suicide, 0.25), procedural ('how to' suicide query, 0.24) and symptoms or diagnosis-related queries (mentions of anxiety, 0.27) co-occurred with an increase in mental health patient deaths. Two more significant negative cross-correlation coefficients, one for non-specific suicide-related query (expression of intent to die by suicide, −0.25) and one for a seeking help related query (references suicide prevention charity, −0.25), were present at lag −3. This indicates that an increase in search volume for these terms was followed by a decrease in mental health patient deaths three months later. Positive significant cross-correlations in the sensitivity period between search volumes and general population suicides were present at lag 0 for a characteristic of suicide (minimum pain suicide, 0.25) and a help-seeking query (generic suicide support, 0.25) as well as at lag 1 for a non-specific suicide related query (generic suicide methods, 0.25) and a help-seeking query (expression of desire to stop being suicidal, 0.27). Finally, there was one negative cross-correlation between the search volume for a non-specific suicide-related search term and suicides in the general population at lag −3 (expression of intent to die by suicide, −0.31).

## Discussion

The present study provides an exploration of the relationship between suicide-related internet search volumes and both mental health patient and general population suicide deaths, with a focus on the types of queries, their temporal dynamics, and implications for online suicide prevention. The findings reveal a small number of statistically significant cross-correlations between suicide-related search volumes and mental health patient and general population suicides. This is consistent with recent literature suggesting that while internet search data can reflect individual distress and population-level trends, the predictive power for trends in the number of suicide deaths is often modest and context-dependent [5,28]. Additionally, the small number of significant associations may suggest that, over time, online prevention efforts such as algorithmic interventions and content moderation may have become more effective.

No pro-suicide queries showed significant associations with suicide deaths. This finding may be explained by long-standing online prevention efforts in the UK, such as Google's inclusion of banners with crisis helplines information in 2010 [29]. These efforts have increased further by the introduction of the UK Online Safety Act in 2023, pressuring internet providers and search engines to block or limit access to pro-suicide websites [30]. Contrary to this finding, a study in the United States found a small but significant association between an increase in pro-suicide web searches and an increase in deaths by self-poisoning [31], which could be explained by the lack of similar online prevention efforts in the United States [32].

Previous research has shown that the temporal associations between suicide-related search volumes and actual deaths by suicide are often unstable and can change depending on the time window analysed [7,9]. As a result, findings

consistent across periods are rare and likely reflect more stable, meaningful behaviours. Sensitivity analysis showed that search volumes of only two queries had a significant cross-correlational coefficient of the same direction and similar size on the same lag (lag 0) in both the full and sensitivity period. One was the non-specific suicide-related query reflecting suicide ideation for mental health patient deaths by suicide and the other was a help-seeking query reflecting seeking suicide support for general population suicides. Notably, these cross-correlational coefficients have some of the highest effect sizes in the present study, suggesting heightened suicidal ideation among mental health patients and a possible unmet need for suicide support among individuals in the general population who are not in contact with mental health services.

A notable finding is that the volumes of two search terms pertaining to the same desired characteristic of suicide (death being pain-free) had significant positive concurrent cross-correlations for both periods and for both populations, meaning that the increase in search volumes for these terms coincided with the increase in deaths by suicide. Variations on these search terms were also examined in previous research and found no significant cross-correlation with actual suicides in the United States and Spain, respectively [7,9,33–35]. This finding might stem from the variations in phrasing, as even subtle differences in how search terms are defined or grouped can affect results [7]. However, in this study, the search terms were analysed specifically as phrased by mental health patients themselves thereby increasing the ecological validity of the findings. Additionally, differences in population and cultural context could contribute to the observed association in the UK but not in other countries, suggesting a potential link between online searches for pain-free ways to die and suicide deaths in both patient and general populations in the UK. This has a broader implication for the importance of ways certain methods are discussed online, namely, the danger of labelling certain methods as "pain-free" and the importance of online moderators actively challenging such claims when they appear in suicide-related online discussions.

An increase in search volume for an anxiety-related query was significantly associated with an increase in mental health patient suicide deaths between 2011 and 2020. Surprisingly, this was the only search term pertaining to symptoms or diagnosis of mental illness that mental health patients reported searching for. In a study by Lee [7] an increase in Google searches for the terms "generalized anxiety disorder" or "anxiety disorder" preceded an increase in suicide deaths by three months. An Irish study observed a small, non-significant negative correlation between the volume of searches for the term "anxiety" and actual suicides [36]. The other significant concurrent cross-correlation was found between a procedural query on how to die by suicide and mental health patients' suicides. Procedural or 'how to' queries have commonly been found to correlate with actual suicide deaths [9,35,37]. One explanation for this finding is that the 'how to' phrasing of the query reflects higher suicidal intent [37]. Furthermore, procedural queries may also indicate a point where individuals are actively considering and preparing for suicide, rather than just experiencing ideation.

There were two terms with significant cross-correlations that concerned specific suicide methods, both in the full study period. However, the direction and timing of the cross-correlations differed by both population (mental health patients v. general population) and method (self-poisoning v. hanging). An increase in search volume related to hanging coincided with an increase in general population suicides. This might be because hanging is the most common suicide method in the general population in the UK [14]. The prominence of this method may therefore influence both the association between search trends for hanging and overall suicide rates. Conversely, searching for a term related to self-poisoning seemed to be protective for mental health patients; the increase in searching for this term preceded a decrease in mental health patients' suicide by two months. For mental health patients, searching for methods may reflect a crisis point that prompts engagement with mental health services, thereby reducing suicide risk in the following months. Furthermore, there have been significant public health interventions in the UK to restrict access to common self-poisoning means [38], which may have reduced the lethality or feasibility of self-poisoning deaths; conversely, similar interventions do not exist for hanging in general population settings.

A search term concerning a phrase communicating intent to die/kill oneself was found to be protective, with an increase in search volume for this phrase preceding a decrease in deaths by suicide in both mental health patients and the general population by three months in the shortened period. Even though this finding may seem counterintuitive, it

might again suggest the effectiveness of the algorithmic moderation of content. If users searching for these phrases are shown preventative resources and subsequently seek support, this could lead to a reduction in suicide deaths in the following months. The final protective term concerned the biggest UK suicide prevention charity, where an increase in searches preceded a decrease in suicide deaths in mental health patients by three months. This is an encouraging result as it may imply that the utilisation of the crisis helpline aided in preventing some suicides, as has been noted in previous research [39].

There are positive cross-correlations on positive lags between search volumes and suicide deaths only in the general population. Positive cross-correlations on positive lags indicate that an increase in suicide volumes precedes an increase in the search volume of suicide-related search terms. In current study, these were present for the two non-specific ssuicide-related queries and a seeking help search term. The non-specific suicide-related queries concerned the word "suicide" and the phrase "suicide methods". Similarly, the seeking help search term implied looking for resources for coping with one's own or others' suicidal thoughts or urges. This pattern suggests that increases in suicide deaths may heighten public concern or awareness, prompting more people to seek information or support online. Such effects are consistent with the Werther effect, where increased media coverage and public discussion following suicides can drive collective attention and online activity towards suicide [9,40]. Additionally, the observed increase in help-seeking searches may reflect individuals' efforts to cope with loss, support others, or address their own distress in the wake of increased suicide rates. This underlines the complex relationship between actual suicide events and online information-seeking behaviours.

It is important to interpret these results in the context of characteristics of the mental health patients who provided the search terms for which Google Trends data was obtained. This subgroup was predominantly young, female, and included a high proportion of LGBTQ+ individuals; these demographics are known for higher rates of digital engagement and help-seeking behaviour online [41]. Most had affective or anxiety disorders and experienced frequent suicidal thoughts. These characteristics may have amplified the observed associations between certain search volumes, particularly those related to help-seeking or suicide methods, and actual suicides. Two differences were found between these participants and other survey respondents (i.e., suicidal mental health patients who engaged in SRIU); participants who provided search terms planned their suicide attempts more and were more likely to search for suicide methods. The higher degree of planning among these patients may mean they are more likely to encounter both risks (e.g., exposure to harmful content) and opportunities (e.g., exposure to online interventions) during their online searches. This underlines the importance of tailoring digital suicide prevention strategies to target those at highest risk, particularly individuals who are actively planning and seeking method information, by ensuring that supportive resources are highly visible and accessible in response to relevant search queries.

## Strengths and limitations

To the authors' knowledge, this is the first study to [1] examine associations between search volume of suicide-related terms and mental health patient deaths by suicide and [2] use search terms provided by the people who actually engaged in SRIU rather than search terms generated a priori by the researchers. Mental health patients, however, are at significantly higher risk for suicide than the general population [42], and their behaviours and needs may differ substantially. Furthermore, by analysing search terms provided by people who actually engaged in suicide-related internet use (SRIU), it was ensured that the data reflected the real language, intentions, and lived experiences of individuals at risk. It also allowed for the identification of terms that may be relevant but are overlooked in traditional research designs. This is particularly important because the effectiveness of online interventions can depend on accurately matching the language and intent of users in distress [43]. Another methodological strength was the use of robust methodology and analysis. The models were carefully selected, accounting for seasonality, outliers, and autocorrelation and the use of multiple data retrievals and weighted averaging added further methodological rigour.

There are also some limitations to the current study. Firstly, it is important to note that associations do not mean causation; both search activity and suicide deaths may be driven by common underlying or external factors [9]. Secondly, chosen statistical approach, which tested multiple lags per time series, required the use of Bonferroni correction to control for multiple comparisons. This conservative adjustment may have limited the detection of smaller, yet potentially meaningful, effects. [44]. Thirdly, it is acknowledged that using UK-wide search data and suicide time series would be optimal. This was not possible since the available suicide data for Northern Ireland did not include month of death. Recent mental health patient suicide data were partially projected due to delays in legal process and data collection. Although NCISH projections are consistently closely correlated with actual numbers with 0–5% margin of errors [45] and sensitivity analyses restricted to earlier years were conducted, some residual bias cannot be ruled out.

While the decision to use search terms provided directly by participants who engaged in SRIU enhanced ecological validity, the characteristics of participants may have amplified the observed associations between certain search volumes. Therefore, the findings are especially relevant to digitally engaged individuals with frequent suicidal ideation and may be less generalisable to those whose suicidal behaviour is more impulsive or who are less active online.

There are some limitations applicable to all research using Google Trends to explore suicidality. Averaging Google Trends data collected across 15 different days helped minimise random fluctuations; however, it could not completely remove sampling bias or limits to replicability. Google frequently updates its algorithms and moderation practices, including the introduction of protective measures for suicide-related searches and these changes are not transparent to researchers. This means older research may underestimate the role of internet searches in suicidality because fewer people used the internet and/or Google in the past and newer research is complicated by Google's evolving moderation and intervention strategies, which may suppress or redirect certain queries. While this evolution in Google's policies is largely positive, it makes current data less reflective of organic search behaviour and, potentially, suicide behaviour. It is worth noting that, with these changes, the content shown to the user following a suicide-related search are probably different today compared to earlier periods. Also, it is important to note that Google Trends was not developed as a research tool and the algorithms Google uses to process, sample, and present search data are proprietary. Therefore, researchers cannot know how search volumes are sampled, how outliers or anomalies are handled, or how changes to Google's algorithm might affect results over time.

## Conclusion and future directions

A recent paper by Rushendran and Chitra [46] states that without knowing who is searching for suicide-related content and why, it is impossible to reliably infer population-level distress or suicide risk from search volumes alone. This study provided some evidence as to who is searching, but the 'why' is still largely unknown. Even seemingly clear 'how to' queries, when considered motivation-wise, may stem from personal crisis, academic research, professional interest, or even curiosity after media coverage. Therefore, future research in this field should investigate intent as well as the content of the suicide-related searches. Studies could also consider analysing the content elicited by the search term studied, which could provide additional insights into why certain search terms are preventative or risk related. For instance, the information and resources in search results may influence user perceptions and subsequent actions, potentially explaining observed associations. However, Google does not provide access to historical search result pages, meaning that researchers cannot retrospectively analyse what users would have seen at a given point in the past. Consequently, these studies should be designed prospectively and search result pages would need to be systematically captured at regular intervals. While online searching for suicide-related information is most likely a prevalent behaviour, this study corroborates conclusions from Tran and colleagues [9] that Google Trends data is likely not suitable for monitoring and prediction of actual suicide behaviours on a population level. However, this approach can still provide valuable insights and help generate hypotheses on suicide-related internet use, as long as the interpretations are tempered by explicit acknowledgement of its limitations.

## Supporting information

**S1 File. Comparison between participants who did and those who did not provide search terms.**
(DOCX)

## Author contributions

**Conceptualization:** Lana Bojanić.

**Data curation:** Lana Bojanić.

**Formal analysis:** Lana Bojanić.

**Investigation:** Lana Bojanić.

**Methodology:** Lana Bojanić, Saied Ibrahim.

**Supervision:** Isabelle M. Hunt, Sandra Flynn, Pauline Turnbull, Saied Ibrahim.

**Validation:** Sandra Flynn, Saied Ibrahim.

**Writing – original draft:** Lana Bojanić.

**Writing – review & editing:** Lana Bojanić, Isabelle M. Hunt, Sandra Flynn, Pauline Turnbull, Saied Ibrahim.

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
