## [Decision Letter · Decision Letter 0]

26 Dec 2025

Dear Dr. Bojanić,

Thank you for submitting your manuscript to PLOS ONE. After careful consideration, we feel that it has merit but does not fully meet PLOS ONE’s publication criteria as it currently stands. Therefore, we invite you to submit a revised version of the manuscript that addresses the points raised during the review process.

**ACADEMIC EDITOR:** **Please address reviewer comments and resubmit for further consideration for publication.**

We look forward to receiving your revised manuscript.

Kind regards,

Souparno Mitra, M.D.

Academic Editor

PLOS One

Journal Requirements:

2. In your Methods section, please include additional information about your dataset and ensure that you have included a statement specifying whether the collection and analysis method complied with the terms and conditions for the source of the data.

3. In the online submission form, you indicated that [Since the data in question represents search terms used for suicide-related purposes, the data will be available to other researchers by request only.].

4. Please note that you should ensure that each author is linked to an affiliation, and clearly indicate the corresponding author, as per our author instructions. Authors’ affiliations should reflect the institution where the work was done (if authors moved subsequently, you can also list the new affiliation stating “current affiliation:….” as necessary).

5. Please include your full ethics statement in the ‘Methods’ section of your manuscript file, and please make sure that the information provided in your submission is also the same as what will be included in the manuscript. In your statement, please include the full name of the IRB or ethics committee who approved or waived your study, as well as whether or not you obtained informed written or verbal consent. If consent was waived for your study, please include this information in your statement as well.

Reviewers' comments:

Reviewer's Responses to Questions

**Comments to the Author**

1. Is the manuscript technically sound, and do the data support the conclusions?

Reviewer #1: Yes

Reviewer #2: Partly

2. Has the statistical analysis been performed appropriately and rigorously?

Reviewer #1: Yes

Reviewer #2: I Don't Know

3. Have the authors made all data underlying the findings in their manuscript fully available?

Reviewer #1: No

Reviewer #2: Yes

4. Is the manuscript presented in an intelligible fashion and written in standard English?

Reviewer #1: Yes

Reviewer #2: Yes

Reviewer #1: The paper examines the association between suicide-related Google Trends searches and actual suicide deaths in the UK. The strength of their paper is their use of terms from people with lived experience, which I think is very commendable and a major strength. Overall, their statistical methods for the time series analysis are impeccable, their writing is good, and the care they have taken with the Google Trends data is excellent, although I will point to one major flaw. The results and discussion are thorough but also measured, which is another strength of this paper.

The Appendix was unfortunately not included in the review file.

Minor:

P1L21: mental health patients

P4L73-5: Consider changing (1) and (2) so as to distinguish from citation style. Even just 1) and 2), or (a) and (b), etc.

Major:

P6L139–143.

I would recommend that the authors reconsider their strategy here:

The Google Trends API data getting started guide1 explain what GT data are:

“The numbers returned are the probability of a short search-session (few consecutive searches), to satisfy the corresponding term restriction, given it was done in the restricted geography (if such exist) and during the time represented by that data point. This probability is multiplied by 10 Million in order to be more human readable. “These values are then scaled to 100 (and rounded) for the Web data.

The values they are working with, are thus the probability of a term being searched, relative to (i.e., weighted by) the nation, for their time period. As such, weighting by the population percentage actually introduces a distortion into the data. I would, however, go further and recommend (but not insist) that the authors consider redoing this work using API data, which provides the raw (unscaled, not-rounded) probability (x 10,000,000 as stated above). This would immensely increase the value of their paper, and allow a much finer comparison of the terms. I have developed and published methods for extracting the required number of samples in a single day, so the main work required would be applying for an API key.

Next, it is odd that the authors weighted the GT data by the nation population size, but did not weight the suicide numbers, but used the “Actual monthly numbers of mental health patients who died by suicide” (P6L152). They weighted what they should not have weighted, and did not weight what they should have weighted. However, a further note on weighting. They did not define the time point for the population size (I do note they used population percentage, not population size, P6L140). To properly weight the suicide rates, it would be best to obtain monthly population estimates per nation (or, if those are not available, to estimate them from the nearest available data, such as quarterly or annual population sizes) and express the suicides over time as a rate per 100,000.

References

1. Google. Google Extended Trends API for Health - Getting Started Guide. Google;

Reviewer #2: Good work on the study "Modest Google Trends Associations with Mental Health Patient and General Population Deaths by Suicide: A Time-Series Analysis Using Patient-Provided Search Terms". I have attached the suggested review comments to the file attached.

**Do you want your identity to be public for this peer review?** For information about this choice, including consent withdrawal, please see our For information about this choice, including consent withdrawal, please see our Privacy Policy .

Reviewer #1: **Yes:** Jacques Eugene RaubenheimerJacques Eugene Raubenheimer

Reviewer #2: **Yes:** Arun PrasadArun Prasad

---

## [Author Response · Author response to Decision Letter 1]

5 Feb 2026

PONE-D-25-53803

Modest Google Trends Associations with Mental Health Patient and General Population Deaths by Suicide: A Time-Series Analysis Using Patient-Provided Search Terms

Dear Dr Mitra,

Thank you for your interest in the paper and the helpful comments from the reviewers. We have now responded to the comments which we believe has improved the paper, and revisions are highlighted in yellow throughout. The journal requirements have now been addressed and we look forward to hearing from you.

Kind regards,

Lana Bojanić (on behalf of all authors)

Journal Requirements:

https://journals.plos.org/plosone/s/file?id=wjVg/PLOSOne_formatting_sample_main_body.pdf [journals.plos.org] and https://journals.plos.org/plosone/s/file?id=ba62/PLOSOne_formatting_sample_title_authors_affiliations.pdf [track.editorialmanager.com]

Response: The manuscript has now been reformatted to meet the journal’s style requirements.

2. In your Methods section, please include additional information about your dataset and ensure that you have included a statement specifying whether the collection and analysis method complied with the terms and conditions for the source of the data.

Response: The following has now been included at the end of methods section:

“Data availability and compliance

This study analysed three data sources, each with distinct availability and terms:

(i) Participant-generated search terms: Participants who provided search terms were recruited online as part of a cross-sectional anonymous survey on suicide-related internet use of mental health patients between June 1 and December 31, 2023. They provided written informed consent by ticking the box in the online questionnaire hosted on Qualtrics. The search terms and coded categories are available upon request from the corresponding author LB (lana.bojanic-2@manchester.ac.uk). Due to ethical considerations regarding the public dissemination of potentially harmful search terms related to suicidality (24), the full list of search terms is not included in the main text.

(ii) Google Trends data: These time-series data are publicly available via Google Trends (https://trends.google.com) and can be reproduced using the methodology described in the Methods section.

(iii) Suicide mortality data: These data are governed by strict confidentiality regulations and are not available for public distribution. Access to the Office for National Statistics and National Records of Scotland data is subject to request. For more information, see https://www.ons.gov.uk/aboutus/whatwedo/statistics/requestingstatistics (ONS) and https://www.researchdata.scot/accessing-data/begin-an-enquiry (NRS). Access to mental health patient suicide data can be requested via application to the Healthcare Quality Improvement Partnership (https://www.hqip.org.uk/national-programmes/accessing-ncapop-data/).

All authors had full access to all of the data in the study and take responsibility for the integrity of the data and the accuracy of data analysis. All analyses complied with the ethical approvals and data governance frameworks specified by each data source provider.”

3. In the online submission form, you indicated that [Since the data in question represents search terms used for suicide-related purposes, the data will be available to other researchers by request only.].

Response: The raw participant-level data for this study cannot be made publicly available. Our ethics approval from the University of Manchester Research Ethics Committee 1 (ref: 2023-16133-28055) explicitly requires that:

• Data may only be accessed and analysed within the NCISH office and are stored on an isolated network not connected to general university infrastructure.

• Access is restricted to the named research team (LB, IH, SI, SF, PT) involved in this study.

These restrictions were imposed as a condition of the ethical approval due to the sensitive nature of the data (mental health patients with suicidal thoughts/behaviours). Making these data publicly available would breach both our approved protocol and UK GDPR/Data Protection Act 2018 obligations.

The core analysis in this study (the association between search volume of suicide-related terms and suicide mortality) is fully dependent on the participant-generated search term data; therefore, providing access to the mortality data alone would not enable replication of our analysis. Replication would require access to the search term dataset, which is restricted under our ethics approval. Therefore, we request an exemption from public data deposition for these datasets.

4. Please note that you should ensure that each author is linked to an affiliation, and clearly indicate the corresponding author, as per our author instructions. Authors’ affiliations should reflect the institution where the work was done (if authors moved subsequently, you can also list the new affiliation stating “current affiliation:….” as necessary).

Response: I confirm that all authors are linked to an affiliation.

5. Please include your full ethics statement in the ‘Methods’ section of your manuscript file, and please make sure that the information provided in your submission is also the same as what will be included in the manuscript. In your statement, please include the full name of the IRB or ethics committee who approved or waived your study, as well as whether or not you obtained informed written or verbal consent. If consent was waived for your study, please include this information in your statement as well.

Response: The full ethical approval section is now as follows:

“Ethical approval

Ethical approval for collecting and analysing participants' suicide-related search terms was obtained from the University of Manchester Research Ethics Committee 1 on the 3rd of April 2023 (ref: 2023-16133-28055). Suicide data is routinely obtained by the National Confidential Inquiry into Suicide and Safety in Mental Health with approval from the North West – GM South REC (reference: ERP/96/136) and Section 251 Approval under the NHS Act 2006 (originally Section 60 of the Health and Social Care Act 2001) (reference: 23/CAG/0024) for England and Wales and Public Benefit and Privacy Panel for Health and Social Care (HSC-PBPP) (reference: 2021-0114) for Scotland.”

Also, we included the following regarding consent in the data availability and compliance section:

“Participants who provided search terms were recruited online as part of a cross-sectional anonymous survey on suicide-related internet use of mental health patients between June 1 and December 31, 2023. They provided written informed consent by ticking the box in the online questionnaire hosted on Qualtrics.”

Review Comments to the Author

Reviewer #1: The paper examines the association between suicide-related Google Trends searches and actual suicide deaths in the UK. The strength of their paper is their use of terms from people with lived experience, which I think is very commendable and a major strength. Overall, their statistical methods for the time series analysis are impeccable, their writing is good, and the care they have taken with the Google Trends data is excellent, although I will point to one major flaw. The results and discussion are thorough but also measured, which is another strength of this paper.

The Appendix was unfortunately not included in the review file.

Minor:

P1L21: mental health patients

P4L73-5: Consider changing (1) and (2) so as to distinguish from citation style. Even just 1) and 2), or (a) and (b), etc.

Response: Thank you for your kind comments on the paper’s strengths. The appendix (S1 File) is now included and the minor comments above have been corrected.

Major:

P6L139–143.

I would recommend that the authors reconsider their strategy here:

The Google Trends API data getting started guide1 explain what GT data are:

“The numbers returned are the probability of a short search-session (few consecutive searches), to satisfy the corresponding term restriction, given it was done in the restricted geography (if such exist) and during the time represented by that data point. This probability is multiplied by 10 Million in order to be more human readable. “These values are then scaled to 100 (and rounded) for the Web data.

The values they are working with, are thus the probability of a term being searched, relative to (i.e., weighted by) the nation, for their time period. As such, weighting by the population percentage actually introduces a distortion into the data. I would, however, go further and recommend (but not insist) that the authors consider redoing this work using API data, which provides the raw (unscaled, not-rounded) probability (x 10,000,000 as stated above). This would immensely increase the value of their paper, and allow a much finer comparison of the terms. I have developed and published methods for extracting the required number of samples in a single day, so the main work required would be applying for an API key.

Response: Thank you for this interesting comment. After exploring API data and your work in more detail we recognise the methodological advantages of its consistently scaled data. However, the current API provides access only to a rolling window of approximately 5 years which would not align with our suicide data that spans twelve years. We will nonetheless consider API-based methods in future work.

The values they are working with, are thus the probability of a term being searched, relative to (i.e., weighted by) the nation, for their time period. As such, weighting by the population percentage actually introduces a distortion into the data. Next, it is odd that the authors weighted the GT data by the nation population size, but did not weight the suicide numbers, but used the “Actual monthly numbers of mental health patients who died by suicide” (P6L152). They weighted what they should not have weighted, and did not weight what they should have weighted. However, a further note on weighting. They did not define the time point for the population size (I do note they used population percentage, not population size, P6L140). To properly weight the suicide rates, it would be best to obtain monthly population estimates per nation (or, if those are not available, to estimate them from the nearest available data, such as quarterly or annual population sizes) and express the suicides over time as a rate per 100,000.

Response: Thank you for this detailed comment on weighting methodology.

Regarding weighting GT data, we weighted Google Trends data by population percentage when creating the composite "Great British" measure because: (1) we needed to reflect search behaviour proportionally across the three constituent regions (England, Wales, Scotland) in a way that searches from larger populations have correspondingly greater influence on the composite and (2) this ensured the composite GT index was comparable to the available suicide data. We have clarified in the text that population percentages were determined using yearly population estimates for each year from 2011 to 2022.

We used absolute suicide counts because the suicide data already represented the complete Great Britain population of suicide deaths (both in the general population and in mental health patients). We note that suicide counts and suicide rates are highly correlated (r = 0.97) in our dataset, such that the correlation between the weighted GT composite and absolute suicide counts result in the same statistical inference to what we would obtain using rates. This is further strengthened by our application of Bonferroni correction, which ensured that only robust correlations ‘survive’ multiple testing. Any correlations that would have remained significant under rates would also remain significant under counts, given their near-perfect correlation. We have clarified this in the methods section, under the subheading ‘Mental health patient and general population suicide time-series’.

Reviewer #2: Good work on the study "Modest Google Trends Associations with Mental Health Patient and General Population Deaths by Suicide: A Time-Series Analysis Using Patient-Provided Search Terms". I have attached the suggested review comments to the file attached.

Some Limitations of the study that may be relevant are as follows -

1) Google trend data has measurement limitations- Search volumes are based on sampled rather than complete search activity, and repeated data extractions can yield different values. Although averaging data retrieved on 15 separate days reduced random variation, this approach cannot fully eliminate sampling bias or guarantee full replicability.

Response: Thank you for pointing this out. We have added the following sentence to the strengths and limitations section:

“Averaging Google Trends data collected across 15 different days helped minimise random fluctuations; however, it could not completely remove sampling bias or limits to replicability.”

2) Search behavior doesnt necessarily equate to Suicidal intent ot attempt possibility.- The selected search terms may reflect diverse motivations, including academic interest, media exposure, curiosity, or third-party searching, rather than personal suicidal ideation.

Response: We agree with this comment and the following is included in the conclusion and future directions section of the discussion:

“A recent paper by Rushendran and Chitra (43) states that without knowing who is searching for suicide-related content and why, it is impossible to reliably infer population-level distress or suicide risk from search volumes alone. This study provided some evidence as to who is searching, but the 'why' is still largely unknown. Even seemingly clear 'how to' queries, when considered motivation-wise, may stem from personal crisis, academic research, professional interest, or even curiosity after media coverage.”

3) Weighting Google trends data by population size assumes similar internet access, search behavior, and help-seeking practices across England, Wales, and Scotland. This may mask meaningful regional differences in digital behavior or suicide risk.

Response: Thank you for your comment. Given the broadly similar internet infrastructure across the UK (Ofcom Connected Nations 2025 report), online searching behaviour (Ofcom Online Nations 2025 report), and mental health related help-seeking practices (Wang et al., 2024), we do not expect meaningful regional differences between England, Scotland, and Wales. We have added the following in the Obtaining and preparing Google Trends data subsection of the Methods section: “Weighting Google Trends data by population size was appropriate given the broadly similar internet infrastructure across the UK (16), online searching behaviour (17), and mental health related help-seeking practices (18).”

Ref: https://www.ofcom.org.uk/siteassets/resources/documents/research-and-data/multi-sector/infrastructure-research/connected-nations-2025/connected-nations-uk-report-2025.pdf?v=407947

https://www.ofcom.org.uk/siteassets/resources/documents/research-and-data/online-research/online-nation/2025/online-nations-report-2025.pdf?v=409837

https://doi.org/10.1136/bmjopen-2023-073731

4) Recent mental health patient suicide data were partially projected due to delays in reporting. While sensitivity analyses restricted to earlier years were conducted, residual bias cannot be ruled out.

---

## [Decision Letter · Decision Letter 1]

26 Feb 2026

Modest Google Trends Associations with Mental Health Patient and General Population Deaths by Suicide: A Time-Series Analysis Using Patient-Provided Search Terms

PONE-D-25-53803R1

Dear Dr. Bojanić,

We’re pleased to inform you that your manuscript has been judged scientifically suitable for publication and will be formally accepted for publication once it meets all outstanding technical requirements.

Kind regards,

Souparno Mitra, M.D.

Academic Editor

PLOS One

Additional Editor Comments (optional):

Reviewers' comments:

Reviewer's Responses to Questions

**Comments to the Author**

Reviewer #1: All comments have been addressed

Reviewer #2: All comments have been addressed

2. Is the manuscript technically sound, and do the data support the conclusions?

Reviewer #1: Yes

Reviewer #2: Partly

3. Has the statistical analysis been performed appropriately and rigorously?

Reviewer #1: Yes

Reviewer #2: I Don't Know

4. Have the authors made all data underlying the findings in their manuscript fully available?

Reviewer #1: No

Reviewer #2: Yes

5. Is the manuscript presented in an intelligible fashion and written in standard English?

Reviewer #1: Yes

Reviewer #2: Yes

Reviewer #1: Thank you for the comments. I will point out that the Google Trends Extended for Health API allows access to data from 2004 to the present, not only the last five years. However, since I did not insist that the authors use this data source, I am satisfied with the results as is.

Reviewer #2: Thank you for the reponses to suggested edits on the paper " Modest Google Trends Associations with Mental Health Patient and General Population Deaths by Suicide: A Time-Series Analysis Using Patient-Provided Search Terms."

**Do you want your identity to be public for this peer review?** For information about this choice, including consent withdrawal, please see our For information about this choice, including consent withdrawal, please see our Privacy Policy .

Reviewer #1: **Yes:** Jacques Eugene RaubenheimerJacques Eugene Raubenheimer

Reviewer #2: **Yes:** Arun PrasadArun Prasad

---

## [Editor Report · Acceptance letter]

PONE-D-25-53803R1

PLOS One

Dear Dr. Bojanić,

I'm pleased to inform you that your manuscript has been deemed suitable for publication in PLOS One. Congratulations! Your manuscript is now being handed over to our production team.

Kind regards,

on behalf of

Dr. Souparno Mitra

Academic Editor

PLOS One